


# Variation of deuterium excess in surface waters across a 5000-m elevation gradient in the east-central Himalaya

Katalyn A. Voss[1], Bodo Bookhagen[1,2], Dirk Sachse[3], Oliver A. Chadwick[1]

[1]Department of Geography, University of California, Santa Barbara.
[2]Institute of Earth Environmental Sciences, Potsdam University.
[3]GFZ German Research Center for Geosciences, Section 5.1 Geomorphology.

*Correspondence to*: Katalyn A. Voss (katalynvoss@geog.ucsb.edu)

**Abstract.** The strong elevation gradient of the Himalaya allows investigation of altitude and orographic impacts on precipitation isotope values as captured in river samples. This study provides new high-elevation data along a 5000-m gradient collected from rain, snow, and glacial-sourced surface waters and time-series data from April to October 2016 to differentiate the time-variable contributions of source waters to the Arun River. We find nonlinear trends in $\delta^{18}O$ and $\delta D$ lapse rates driven by samples collected at high elevations and a distinct seasonal signal indicative of moisture source influences the surface-water isotope values. Deuterium excess is correlated to snowpack and used to track melt events during the monsoon. Our analysis identifies contributions from snowpack to river discharge before the monsoon onset followed by a 5-week transition to Indian Summer Monsoon-sourced rainfall around mid-June 2016.

## 1 Introduction

Seasonal snow and glacial melt from the Himalaya support the domestic, industrial, and agricultural water demands of over 1.5 billion people in the South Asia region (Immerzeel et al., 2010; Viviroli et al., 2007). Yet, regional and local accounting of these water resources is sparse despite the immediate relevance for water management and the need to predict future variability in water supply that will result from climate change (Beniston, 2003; National Research Council, 2012). Water stable isotopes and their spatiotemporal variation provide a unique tracer to characterize contributions of different water sources, including snow and glacial melt (Aggarwal et al., 2006; Gat, 2010; Kendall & McDonnell, 2012). Differences in water isotope values emerge from fractionations occurring during water-vapor transport and precipitation, which are correlated to altitude, latitude, precipitation intensity, and distance from the moisture source. In addition, post-deposition fractionation from evaporation in soil and sublimation in snow fields can modify the isotopic composition of surface waters (Dansgaard, 1964; Gat, 2010). In high mountain regions, distinguishing the isotopic values of different water sources that contribute to stream flow is complicated by the extreme effects of orographic rise, seasonal changes in water-vapor sources, melting of snow fields and glaciers, and difficulty to access high-altitude regions.

In the Himalaya, the combination of decreasing temperatures and increasing rainout with increasing elevation accentuates fractionation processes and results in water depleted in heavy isotopes. This relationship, reported as stable isotope lapse rates, ranges in the Himalaya from -0.45 to -4.4 ‰ km$^{-1}$ for $\delta^{18}O$ and -8.3 to -33.0 ‰ km$^{-1}$ for $\delta D$, but these estimates largely draw on surface water samples from streams in low and mid elevations less than 5000 m (Hren et al., 2009; Meese et al., 2018; Poage & Chamberlain, 2001; Racoviteanu et al., 2013; van der Veen et



al., 2018; Varay et al., 2017; Wilson et al., 2015).  High-elevation samples are sparse but essential to accurately characterize mountain-sourced surface waters, especially in regions with significant contributions from isotopically depleted snow and glacial melt.

5         In addition to a strong elevation control, precipitation in the Himalaya is sourced from two moisture regimes – the Indian Summer Monsoon (ISM) and Winter Westerly Disturbances (WWD) – each of which has a distinct isotopic signature due to its source, path, and precipitation intensity. The ISM originates in the Bay of Bengal and Indian Ocean and moves across the Indian subcontinent before intersecting with the orographic barrier of Himalaya (Bookhagen and Burbank, 2010).  WWD moisture originates in the Mediterranean, Black, and Caspian Seas and

follow a continental path across the Middle East and Central Asia before reaching the western edge of the Himalaya (Barry, 2008; Cannon et al., 2015; Lang & Barros, 2004). The bulk of ISM precipitation falls as rain from June to September while WWD precipitation is deposited as snowpack from December to April (Barry, 2008; Bookhagen and Burbank, 2010; Lang and Barros, 2004). Deuterium excess (d-excess), which is derived as a deviation from the global

meteoric water line (GMWL), is a tool to measure kinetic fractionation effects related to humidity, moisture recycling, and post-deposition processes, including sublimation (Dansgaard, 1964; Froehlich et al., 2002). In the Himalaya, d-excess can be used to separate ISM- and WWD-sourced precipitation because the ISM is marked by low d-excess values while WWD have relatively high d-excess values (Balestrini et al., 2016; Hren et al., 2009; Jeelani et al., 2013;

Pande et al., 2000).  Previous studies largely sampled in the post-monsoon season and reflect a dominant ISM moisture source, and little is known about the temporal evolution of surface water isotope values driven by seasonal moisture source variation, particularly the influence of seasonal snowmelt (Grujic et al., 2018; Hren et al., 2009; Meese et al., 2018; Varay et al., 2017; van der Veen et al., 2018).

25         This study, based in the Arun River basin in eastern Nepal, captures the temporal and spatial variability in water stable isotope values driven by seasonal ISM and WWD moisture sources with data from both the pre- and post-monsoon that include surface water samples from drainages sourced from rain, snow, and glacial melt. It contributes to a growing understanding of the sources and timing of water contributions at low and mid elevations and offers isotopic data

for surface water at elevations above 5000 m, where few samples have been collected previously. In the past, models of isotope depletion at high elevation relied on linear regression models to extrapolate isotope values from low and mid elevations. We show that there are strong nonlinear changes in the isotopic values of water contributions to the Arun River driven by high-elevation surface water sourced from snow and glacial melt dominated drainages.

**2 Study Site**

        The Arun Valley receives precipitation from both the ISM and WWD and exhibits one of the steepest elevation gradients in the eastern Himalaya (200 m to 8480 m).  The 33,000-km$^2$ Arun River contains ~15% glacierized area and is estimated to receive ~25% of annual streamflow from WWD-sourced snowmelt and ~70% from ISM-sourced rainfall (Bookhagen &

Burbank, 2010; RGI Consortium, 2017). This study includes data from two sub-basins of the Arun River: the Barun Khola and the Sabha Khola (Fig. 1).  The Barun Khola is a 468-km$^2$ watershed with a mean catchment elevation of 4758 masl and ~30% glaciated area (Fig. 1a). The confluence of the Barun Khola with the Arun River is at Barun Bazaar.  The Sabha Khola is a 549-km$^2$ nonglacierized watershed with a mean elevation of 1503 masl (Fig. 1b).  The

confluence of the Sabha Khola and Arun River is located near Tumlingtar.  Combined, the Barun





and Sabha Khola span from 200 m to 8480 m at the summit of Mt. Makalu. The relief of the region allows for dense sampling over a wide range of elevations while minimizing changes in latitude and longitude. This study investigates the elevation fingerprint on water stable isotope values along an elevation gradient of nearly 6000 m and explores potential applications of water stable isotopes to parse surface water sourced from ISM versus WWD precipitation.

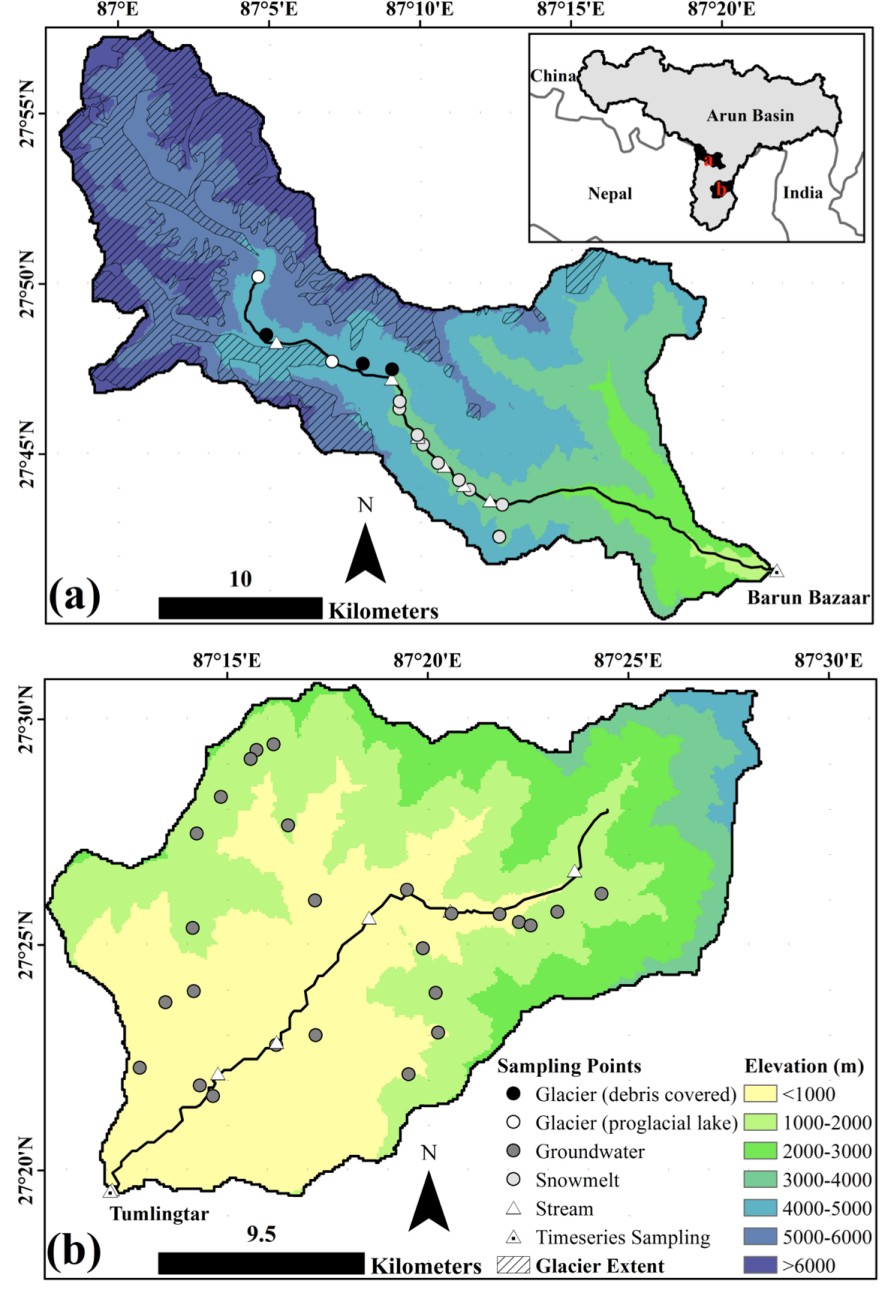



**Figure 1.** Map of the Barun Khola and Sabha Khola, two sub-basins of the Arun watershed located in eastern Nepal (see map inset).  Glacier extent is from the Randolf Glacier Inventory derived from Global Land Ice Measurements from Space (GLIMS) (RGI Consortium, 2017).  Synoptic samples were collected in the Barun and Sabha Kholas (a and b, respectively) in the pre- and post-monsoon seasons.  Time-series samples were collected at the confluence of the Barun and Sabha Khola with the Arun River at Barun Bazaar and Tumlingtar, as labeled.  The dominant water source for each drainage area is indicated by the shading of the circles.  Main stem stream samples are indicated with a triangle.

## 3 Data and Methods

Following the methodology of hydrograph studies in the Andes and Himalaya (Baraer et al., 2009; Mark et al., 2005; Wilson et al., 2015), a synoptic sampling approach was used to characterize spatial variation in stable isotope values in the Sabha and Barun Kholas during two field campaigns.  A pre-monsoon field campaign was conducted from April to May 2016 with repeat sampling after the monsoon during October and November 2016 (see data repository for list of samples). The shoulder periods of the monsoon were targeted to minimize influence from recent precipitation events.  When possible, stream samples were collected from the main stem of the Barun Khola and Sabha Khola every 200-m of elevation gain (N = 28; 14/season).  Samples were also collected from surface water drainages that contained extensive snowpack (N = 22; 11/season), proglacial lakes (N = 4; 2/season), debris-covered glaciers (N=6; 3/season), as well as springs and surface-water tributaries draining from forested and agriculture-dominated areas (N = 50; 27/season).  Local Nepali schoolteachers were trained in water sampling techniques and collected time-series data at the confluence of the Barun and Sabha Kholas with the Arun River at Barun Bazaar and Tumlingtar, respectively.  Samples were collected every 4 days from late April/early May through October 2016 (N= 41 for the Sabha Khola; N=45 for Barun Khola).

Two sets of water samples were collected for each site.  Each sample was filtered through 0.45-μm glass filters into 30-mL polyethylene Nalgene bottles that were rinsed three times with filtered water.  Bottles were filled completely to produce a positive meniscus, sealed tightly, and wrapped with tape to prevent the formation of air bubbles and possible evaporation before laboratory analysis.  Bottles were stored in a dark place until the completion of the field season and then stored in a refrigerator at 4°C at UC Santa Barbara.  During the pre- and post-monsoon synoptic sampling campaigns, field duplicates were collected every 10 samples.

Samples were analyzed for oxygen and hydrogen stable isotopes ($\delta^{18}$O and deuterium ($\delta^2$H or $\delta$D)) at the German Research Center for Geosciences (GFZ) Organic Surface Geochemistry Lab with a Picarro L-2140i Ringdown Spectrometer.  Isotope values for $\delta^{18}$O, $\delta$D, and d-excess are reported in delta notation as parts per thousand as related to their deviation from Vienna Standard Mean Ocean Water (VSMOW).  Precision for $\delta^{18}$O and $\delta$D measurements was ±0.03 $^0/_{00}$ and ±0.33 $^0/_{00}$, respectively.  Analytical uncertainties for the water stable isotope measurements are less than 1% for all samples and are reported as standard deviation in the data repository.  D-excess was calculated for each sample as deviation from the Global Meteoric Water Line (GMWL) where d-excess = $\delta$D - 8*$\delta^{18}$O (Dansgaard, 1964).  Data for individual samples are listed in Table 1.

To supplement the isotope interpretation, snow extent was calculated for the pre- and post-monsoon seasons. Two Landsat 8 OLI images with minimal cloud cover were selected from the beginning (March 24th, 2016) and end (November 11th, 2016) of the synoptic sampling



campaigns. Regions of interest were selected for each image to conduct a supervised classification using Maximum Likelihood classification. The classification output was validated and adjusted to ensure accurate representation of snow extent. Weekly averaged land surface temperatures for the Sabha Khola and Barun Khola were calculated from the MODIS/Terra Land

Surface Temperature Daily Global 1-km gridded data (MOD11A1) product for the duration of the time-series data (April through November 2016) to provide further insight to climate processes during the study period (Wan, 2015). Days with extensive cloud cover where less than 10% of pixels in the study basins were visible are excluded from the analysis. Daily rainfall data were averaged over the Barun Khola and Sahba Khola basins with precipitation data from the

U.S. Geological Survey and UC Santa Barbara Climate Hazards Group Infrared Precipitation with Stations (CHIRPS) dataset (Funk et al., 2015).

## 4 Results

### 4.1 Water stable isotope values and lapse rates

Water stable isotope values in the Barun Khola and Sabha Khola show a distinct

elevation fingerprint with a clear depletion effect that strengthens at higher elevations (Fig. 2). In both the Barun and Sabha Kholas, $\delta^{18}O$ and $\delta D$ values in the post-monsoon season are depleted in heavy isotopes relative to the pre-monsoon season. On average, water samples collected in the Sabha Khola watershed are notably enriched relative to the Barun Khola (see data respository). In the Barun Khola, the post-monsoon $\delta^{18}O$ and $\delta D$ stable isotope lapse rates (-

2.8±0.4 $^0/_{00}$ km$^{-1}$ and -23.1±3.5 $^0/_{00}$ km$^{-1}$, respectively) are weaker than the pre-monsoon signal (-5.8±0.6 $^0/_{00}$ km$^{-1}$ and -49.2±5.2 $^0/_{00}$ km$^{-1}$). In contrast, the pre-monsoon $\delta^{18}O$ and $\delta D$ stable isotope lapse rates in the Sabha Khola (-0.5±0.2 $^0/_{00}$ km$^{-1}$ and -1.0±1.3 $^0/_{00}$ km$^{-1}$) are weaker than the post-monsoon lapse rates (-1.2±0.2 $^0/_{00}$ km$^{-1}$ and -7.7±1.1 $^0/_{00}$ km$^{-1}$). The gap from 3000-4000 m is a consequence of inaccessible terrain in the lower reaches of the Barun Khola. When

all samples are combined, the stable isotope lapse rates for the $\delta^{18}O$ and $\delta D$ are -1.6±0.2 $^0/_{00}$ km$^{-1}$ and -11.6±1.4 $^0/_{00}$ km$^{-1}$ in the pre-monsoon seasons and -2.1±0.1 $^0/_{00}$ km$^{-1}$ and -16.7±0.6 $^0/_{00}$ km$^{-1}$ in the post-monsoon season. While all of the isotope lapse rate regressions are statistically significant (p < 0.01), the coefficient of determination ($r^2$) of the ordinary least squares (OLS) regressions for the Sabha Khola are relatively weaker (0.01-0.59) than the Barun Khola (0.66-

0.81). The lower $r^2$ values in the Sabha Khola are indicative of greater variability in the low-elevation samples, which reflect increased evaporation from terraced agriculture fields and mixing with subsurface groundwater (additional weighted regression results are reported in Table 3).












**Figure 2.** Comparison of pre- and post-monsoon $\delta^{18}$O, $\delta$D, and d-excess values (a, b, and c, respectively) in the Barun Khola and Sabha Khola relative to the mean catchment elevation of the drainage areas of the samples. OLS regressions represent the stable isotope lapse rates for each kilometer of elevation gain. Analytical uncertainty of stable isotope measurements (y-error bars) are less than 1% and smaller than data points. 95% confidence intervals of the OLS regressions are indicated with dotted lines.

D-excess values also exhibit a distinct elevation relationship (Fig. 2c). They are positively correlated in mean catchment elevations up to ~3000 m and then switch to a negative correlation after ~4000 m. In the pre-monsoon season, the deuterium excess lapse rate is -2.8±0.4 $^0/_{00}$ km$^{-1}$ for the Barun Khola samples with mean catchment elevations ranging from 4000 m to 6000 m and 3.4±0.5 $^0/_{00}$ km$^{-1}$ in the Sabha Khola samples with mean catchment elevations between 350 m and 3000 m. In the post-monsoon season, the elevation influence weakens to -0.8±0.2 $^0/_{00}$ km$^{-1}$ in higher elevation catchments and 2.2±0.5 $^0/_{00}$ km$^{-1}$ in lower elevation catchments. Across seasons, the slope of the d-excess versus elevation relationship is roughly the same in the low-elevation regions; however, it weakens in the high-elevation regions. Similar to the $\delta^{18}$O and $\delta$D data, there is greater variability in the low-elevation regions.

To further elucidate the relationship between precipitation moisture source and d-excess, we compare the change in snow-covered area of the drainage basins for samples in the pre- and post-monsoon seasons with the difference in d-excess across seasons (Fig. 3). There is a clear relationship between loss of snowpack and decreased d-excess values from the pre- to post-monsoon seasons. The anomaly is a sample draining from a recent landslide scar, which may indicate rapid subsurface flow and contribution from other sources, for example groundwater.

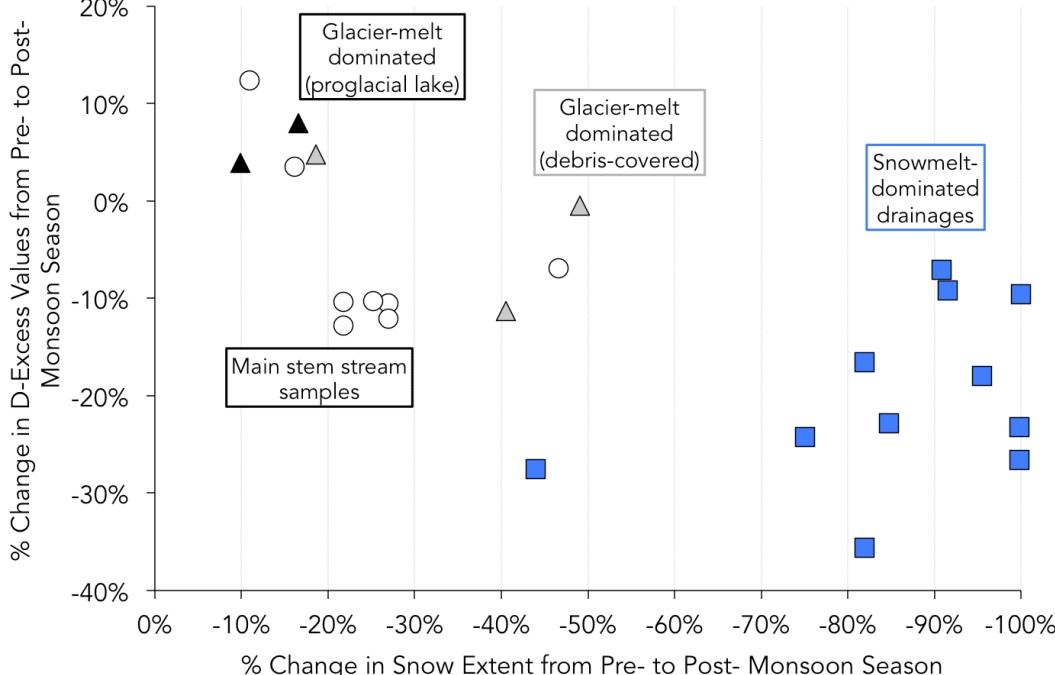



**Figure 3.** Comparison of the change in snow covered area with d-excess values for samples collected in the pre- and post-monsoon seasons. Change in snow was calculated based on the difference in snow extent from Landsat 8 OLI imagery between 24 March and 11 November 2016 for the contributing drainage areas of each sample. Samples are classified based on the dominant water source type for the drainage area in the pre-monsoon season.

## 4.2 Temporal variation in deuterium excess

D-excess exhibits a distinct temporal signal from late April to October in both the Barun Khola and Sabha Khola (Fig. 4). From late April to early June, during the onset of the monsoon, d-excess values increase until 4 June 2016 at which point they decline until 14 July 2016 and stabilize for the duration of the time series. D-excess values in the Sabha Khola are lower than the Barun Khola throughout the time series with the exception of a large drop in d-excess values in the Barun Khola in early September. Temperatures in the Barun Khola, which best represent high-elevation regions where snowmelt may be occurring, are relatively high and stable until early June, then decrease until mid July with a sudden increase in temperatures in late August. The surface temperature declines through September and remains stable, but relatively low, until mid October. The rainfall data indicate sparse rainfall during May and more continuous rainfall from June to September that then diminish in October.







**Figure 4**. Temporal variation of week-averaged land surface temperature, rainfall, and d-excess values sampled at the mouth of the Barun Khola and Sabha Khola from late April through October 2016. Temperature data are week-averaged daily land surface temperature values from MOD11A1 product from the Barun Khola. Rainfall data from CHIRPS are averaged over the Barun and Sabha Khola basins, respectively. D-excess values in the Barun and Sabha Kholas, increase through May, peak in June and decline through September. The peaks in May and June represent an increasing contribution of WWD-sourced snow from melting at the onset of the monsoon (higher d-excess values) followed by a transition to ISM-sourced rainfall (lower d-excess values) during the summer season. The decline in d-excess in the Barun Khola in early September is indicative of a glacial melt pulse following peak temperatures in August (see bottom panel). These events are marked with shaded boxes and labeled accordingly.

## 5 Discussion

### 5.1 Isotopic lapse rates strongly controlled by elevation and water sources

Through targeted sampling in a low- and high-elevation watershed replicated across the pre- and post-monsoon seasons, our study indicates nonlinear trends in $\delta^{18}O$ and $\delta D$ stable isotope lapse rates that are strongly controlled by elevation and seasonal moisture source. Given the terrain of the region, samples collected in the Barun Khola encompassed a nearly 4-km vertical range over a 33-km horizontal path while Sabha Khola samples covered a 1.2-km elevation rise within a 25-km path. This dense sampling approach minimizes the influence of variables other than elevation. That said, when we compare the pre- and post-monsoon data, there is evidence that seasonal variation in moisture source strongly influences stable isotope lapse rates. The weakening stable isotope lapse rate from the pre- to post-monsoon season in the Barun Khola is driven by decreased variability and depletion of $\delta^{18}O$ and $\delta D$ values from samples draining from basins with mean elevations between 4000 m and 5000 m (Fig. 2). These catchments are dominated by WWD snowpack in the spring and transition to ISM-sourced water in the fall after most or all snowpack has melted (Smith and Bookhagen, 2018). Glacier melt-dominated samples, which do not undergo a significant moisture source change, exhibit less variance from the pre- to post-monsoon seasons. In the lower-elevation sites, the stable isotope lapse rates increase from the pre- to post-monsoon season, which may reflect a transition from subsurface flow that underwent evaporation-driven fractionation in the pre-monsoon season to recent elevation-controlled ISM rainfall inputs in the post-monsoon season.

Our isotope lapse rates most closely align with van der Veen et al., (2018), whose data also exhibit nonlinear $\delta D$ lapse rates that strengthen around 4000-m elevation, and Racoviteanu et al., (2013), whose study includes high-elevation snow samples from the eastern Himalaya (Fig. 5 and details in Table 2). Varay et al., (2017) demonstrate a strengthening of isotopic lapse rates from the pre- to post-monsoon season in the low-elevation Tons and Yamuna watershed while the isotopic lapse rate in the higher elevation Sutlej basin remains relatively stable across seasons. In general, studies from the west and central Himalaya report stable isotope lapse rates that are notably weaker than those found in our study. We attribute the differences in stable isotope lapse rates to the paucity of high-elevation data, the spatial coverage of our sampling approach, as well as the increased contribution of $\delta^{18}O$- and $\delta D$-depleted glacier and snowmelt to surface water at high-elevation sites. These factors emerge as clear controls on $\delta^{18}O$ and $\delta D$ stable isotope lapse rates when we compare studies across the Himalayan front (Fig. 5). Indeed, in their review article, Poage & Chamberlain, (2001) report average $\delta^{18}O$ stable isotope lapse rates of -4.1 $^0/_{00}$ km$^{-1}$ from eight studies in high-altitude regions (including the Himalaya);





however, the authors note that the $r^2$ values of the regressions from these studies is relatively weak as a result of greater scatter and variability in the data, which the authors attribute to complex precipitation patterns and post-depositional processes in snowpack. Our data reinforce that δ18O- and δD water stable isotope lapse rates exhibit nonlinear trends that increase at high elevations.

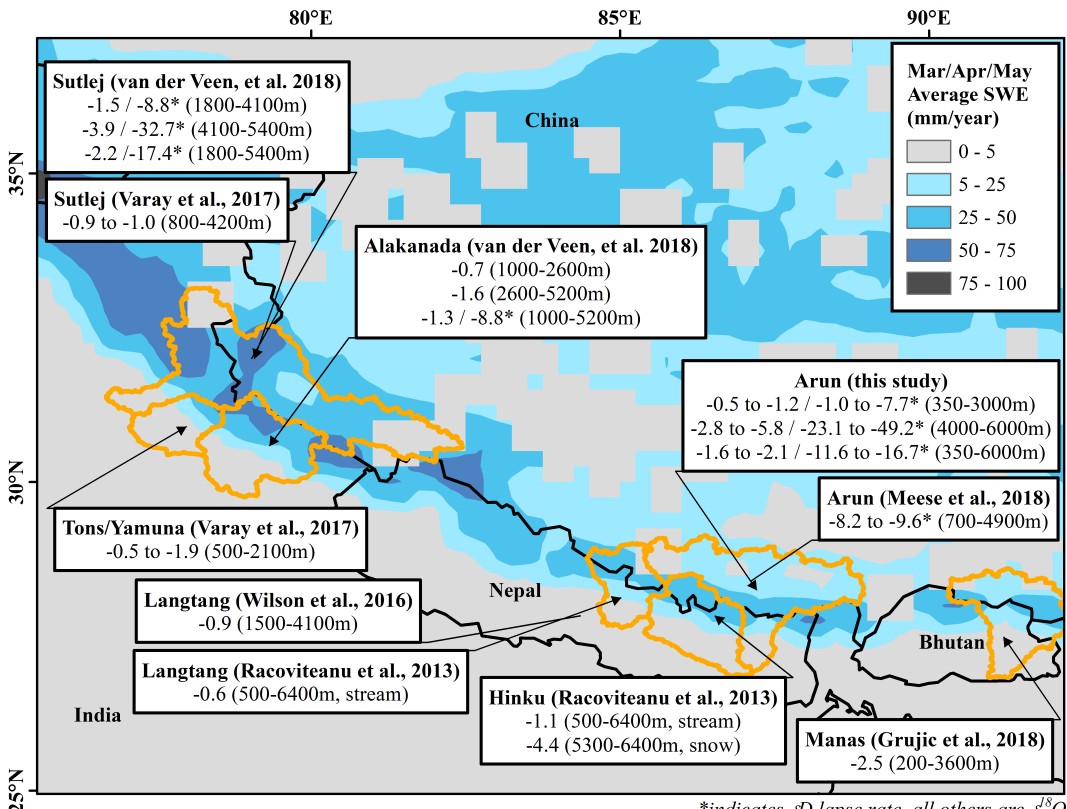

**Figure 5.** Comparison of reported isotope lapse rates across major watersheds in the Himalaya and average snow water equivalent (SWE) in the pre-monsoon (March, April, May) season from 1987 to 2009. Isotope lapse rates are referenced accordingly with additional information in Table 2. SWE averages are derived from passive microwave data from the special sensor microwave imager as reported in Smith & Bookhagen, (2018) and courtesy of T. Smith.

### 5.2 Deuterium excess as a tracer for seasonal moisture source transitions

Deuterium excess emerges as a useful tool to investigate the variation in water stable isotope values driven by precipitation patterns and seasonal moisture source transitions. The d-excess lapse rate (Fig. 2) may reflect two distinct processes: (1) increased evaporation and moisture recycling in lower elevations and (2) post-depositional processes, such as sublimation, in the higher elevation regions (Smith and Bookhagen, 2018). Average d-excess values for WWD-sourced water are relatively high (>15 $^0/_{00}$) whereas ISM-sourced water averages are lower (~8-12 $^0/_{00}$) (Gat & Carmi, 1970; Hren et al., 2009). The pre-monsoon samples in the Barun Khola that drain snowmelt-dominated basins (mean catchment elevations between 4000-





5000 m) align with average WWD d-excess values whereas the post-monsoon samples more closely resemble the ISM, indicating a distinct transition and mixing of moisture sources across seasons (Jeelani et al., 2013). Furthermore, the melting and refreezing of snow accentuates the elevated d-excess values found in drainage basins dominated by snowmelt as compared to basins
dominated by glacial melt (Cooper, 1998; Taylor et al., 2002). As expected, the majority of d-excess values from samples collected in the Sabha Khola in both the pre- and post-monsoon seasons fall within the average range of ISM d-excess values with the exception of stream samples collected from the headwaters of the Sabha Khola, which have snowpack from WWD precipitation and exhibit elevated d-excess values compared to the rest of the basin.
10          There is a clear correlation between loss of snowpack from pre- to post-monsoon season and decreased d-excess values (Fig. 3), which provides further evidence of a transition in moisture sourced from WWD-derived snowpack (with relatively high d-excess values) in the pre-monsoon season to ISM-derived rainfall in the post-monsoon season (with relatively lower d-excess values). Samples from the main stem of the river and glacier melt-dominated drainage
basins exhibit less variation in d-excess values across seasons, reinforcing our hypothesis that high d-excess values are an indicator of WWD-sourced snowpack in the region. These results complement previous studies across the Himalaya that found similar patterns in d-excess linked to precipitation moisture sources in the Himalaya. In a broad survey across the Himalaya and Tibetan Plateau, Hren et al., (2009) notes that d-excess values are strongly correlated to elevation
and longitude, providing a distinct tracer for water vapor transported from the Bay of Bengal, Indian Ocean, and westerly-derived continental sources. Van der Veen et al., (2018) notes a transition in d-excess lapse rates at ~4200 m in the Sutlej (a shift from ~1.5 $^{0}/_{00}$ km$^{-1}$ below 4200 m to -3.0 $^{0}/_{00}$ km$^{-1}$ above 4200 m) that correlates to increased snowpack in the high-elevation regions of the watershed. The d-excess breakpoint does not exist in the Alaknanda basin, which
receives less snowpack and is a lower-elevation catchment. Considered in tandem with the spatial variability of WWD-derived snowpack across the Himalaya (Fig. 5), these results predict increased seasonal variation in d-excess and isotope values as we move westward along the Himalayan front.
            Building from the results of our synoptic sampling campaign, the time-series data (Fig. 4)
show increasing d-excess values at the onset of the monsoon, indicating increased contributions from high d-excess, WWD-derived snowpack associated with surface temperatures greater than 0°C. Snow melt dominates the hydrograph until early June at which point the hydrograph transitions to lower d-excess, ISM-sourced rainfall, eventually stabilizing around mid-July when ISM rainfall dominates the water supply in both catchments (as indicated with an arrow in the
middle panel of Fig. 4). Both the Barun and Sabha Khola time-series data include peaks in d-excess at the onset of the monsoon in May and June (Fig. 4). Two processes could explain these peaks. First, the rivers may be flushed with snowmelt at the onset of the monsoon driven by a combination of increasing temperature and rain-on-snow melting events (Smith and Bookhagen, 2018). Second, the peaks could indicate mixing with pre-monsoon rainfall events which, based
on Balestrini et al., (2016)'s study in the Khumbu region, may have elevated d-excess values. This is particularly true in the Sabha Khola, where d-excess values in early monsoon rainfall events could be further elevated due to high humidity, increased evaporation, and moisture recycling with local water vapor. Either of these processes alone or a combination of the two could drive the peaks in d-excess at the monsoon onset. The drop in d-excess values in the Barun
Khola in early September is most likely associated with a pulse of low d-excess glacial melt water following high temperatures throughout August.



## 6 Conclusions

Our results indicate that Himalayan surface water stable isotope lapse rates strengthen in
high-elevation regions.  These results have important consequences for modeling water stable
isotopes, particularly in the interpretation of paleoclimate and paleoelevation records, and
reinforce the need for additional high-elevation sampling campaigns from areas with high relief
in the Himalaya and other high mountain regions around the globe.  Additional high-elevation
sampling is necessary to further characterize the nonlinear patterns of water stable isotope lapse
rates.  Our study also indicates that the isotopic fingerprints of high-elevation samples are
controlled by the relative influence of snow and glacial melt as well as regional climate
processes, particularly the ISM and WWD, and as such are temporally variable on interannual
time scales. As evidenced in the time-series data, this renders isotopic tracers valuable in
tracking individual hydro-metrological events (see also Balestrini et al., 2016 in the Khumbu and
van der Veen et al., 2018 in northwest India). Additional studies from other regions of the
Himalaya that capture the variability in the strength of the ISM or WWD and incorporate
sampling sites from snowmelt, glacial melt, groundwater, and rainfall are needed to further
validate the spatiotemporal patterns described here.

This study offers a baseline analysis from which we can predict future changes in water
supply that will arise as glaciers continue to retreat and the intensity and timing of the ISM and
WWD become more variable (Bolch et al., 2012; Cannon et al., 2015; Malik et al., 2016).  In
particular, our ability to identify the timing and magnitude of snowmelt pulses contributing to
streamflow is essential to inform immediate water management decisions related to irrigation
and hydropower as well as evaluate potential risk from natural hazards, especially flooding.
These results contribute to a growing body of work from the Himalaya and Andes (see Hill et al.,
(2018)) that utilize isotopic tracers, particularly deuterium excess, to define the spatiotemporal
indicators of rain, snow and glacial meltwaters, and groundwater to assess seasonal fluxes in
high mountain water resources.

*Data availability.* Isotope data for samples can be found in the data repository associated with
the manuscript.  All remotely-sensed snow extent and rainfall data are publically available at the
URLs provided in the references.

*Author contributions.* KV, BB, and OC devised the study; KV completed all fieldwork and
sample preparation; DS provided lab facilities and conducted isotope data analysis; KV was the
lead author with inputs from all other co-authors.

*Competing interests.* The authors declare that they have no conflicts of interest.

*Acknowledgments.* The authors would like to thank Dr. Christina Tague for her feedback and
comments on the manuscript.  The authors also thank Taylor Smith for providing the regional
snow water equivalent data, William Turner for his help processing the CHIRPS rainfall data,
Oliver Rach for his lab support, and Dr. Dhananjay Regmi and the Himalayan Research
Expedition team for their support during fieldwork.  Special thanks to Shanker Rai, Labina Rai,
Kedar Prasad Rai, Bhuvan Halide Rai, Guman Rai, Jabbar Rai, Chandra Rai, and Bikum Rai for
their work as field assistants. Funding for the study was provided by generous support from the
National Science Foundation Graduate Research Fellowship Program grant number DGE





1144085, the Robert and Patricia Switzer Foundation, the American Geophysical Union Horton
Hydrology Research Grant, P.E.O. International, the UCSB Department of Geography, and
UCSB Broom Center for Demography. D. Sachse was partially funded by an ERC Consolidator
Grant (STEEPclim) (Grant agreement No. 647035).

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





| Sample Name | Basin | Dominant Source Water for Drainage Area | Mean Catchment Elevation (m) | Pre-monsoon δ18O (0/00) | Pre-monsoon δ18O std dev (0/00) | Pre-monsoon δD (0/00) | Pre-monsoon δD std dev (0/00) | Pre-monsoon D-excess (0/00) | Post-monsoon δ18O (0/00) | Post-monsoon δ18O std dev (0/00) | Post-monsoon δD (0/00) | Post-monsoon δD std dev (0/00) | Post-monsoon D-excess (0/00) | % Change Snow Covered Area |
|---|---|---|---|---|---|---|---|---|---|---|---|---|---|---|
| K1 | Sabha | Groundwater | 1113 | -5.33 | 0.05 | -35.02 | 0.28 | 7.61 | -6.37 | 0.03 | -41.38 | 0.19 | 9.57 | |
| K2 | Sabha | Groundwater | 1113 | -5.66 | 0.01 | -39.44 | 0.13 | 5.83 | -6.38 | 0.04 | -42.92 | 0.06 | 8.14 | |
| K3 | Sabha | Groundwater | 820 | -5.77 | 0.03 | -35.95 | 0.12 | 10.25 | -5.86 | 0.03 | -37.81 | 0.12 | 9.06 | |
| K4 | Sabha | Stream | 1746 | -6.45 | 0.02 | -37.55 | 0.07 | 14.09 | -7.66 | 0.04 | -48.54 | 0.34 | 12.77 | |
| K5 | Sabha | Groundwater | 782 | -5.78 | 0.03 | -37.21 | 0.12 | 9.04 | -6.58 | 0.03 | -44.17 | 0.09 | 8.46 | |
| K6 | Sabha | Stream | 1808 | -6.50 | 0.05 | -37.65 | 0.09 | 14.35 | -7.82 | 0.04 | -50.11 | 0.11 | 12.48 | |
| K7 | Sabha | Groundwater | 782 | -6.31 | 0.03 | -40.44 | 0.11 | 10.04 | -6.29 | 0.02 | -40.73 | 0.12 | 9.58 | |
| K8 | Sabha | Groundwater | 1306 | -6.92 | 0.01 | -43.42 | 0.10 | 11.97 | -7.48 | 0.02 | -47.73 | 0.10 | 12.08 | |
| K9 | Sabha | Groundwater | 1306 | -7.13 | 0.02 | -44.34 | 0.06 | 12.73 | -7.94 | 0.01 | -50.27 | 0.07 | 13.23 | |
| K10 | Sabha | Groundwater | 1521 | -7.85 | 0.03 | -49.18 | 0.07 | 13.61 | -7.69 | 0.02 | -48.13 | 0.09 | 13.43 | |
| K12 | Sabha | Groundwater | 1053 | -7.07 | 0.02 | -44.28 | 0.09 | 12.31 | -7.13 | 0.03 | -45.61 | 0.08 | 11.47 | |
| K13 | Sabha | Stream | 2429 | -7.14 | 0.03 | -41.52 | 0.10 | 15.64 | -8.69 | 0.04 | -56.05 | 0.07 | 13.46 | |
| K14 | Sabha | Groundwater | 1853 | -6.66 | 0.03 | -42.50 | 0.13 | 10.75 | -6.81 | 0.02 | -43.53 | 0.11 | 10.98 | |
| K15 | Sabha | Groundwater | 1853 | -7.25 | 0.05 | -45.07 | 0.09 | 12.91 | -7.54 | 0.03 | -46.66 | 0.09 | 13.66 | |
| K16 | Sabha | Groundwater | 2268 | -7.45 | 0.03 | -45.04 | 0.12 | 14.53 | -8.64 | 0.04 | -54.67 | 0.10 | 14.44 | |
| K17 | Sabha | Groundwater; snowmelt | 2543 | -8.16 | 0.03 | -50.19 | 0.12 | 15.11 | -9.38 | 0.02 | -60.38 | 0.16 | 14.65 | -100% |
| K19 | Sabha | Stream (headwaters) | 2962 | -6.45 | 0.04 | -34.27 | 0.16 | 17.36 | -9.51 | 0.02 | -61.30 | 0.09 | 14.77 | -100% |
| K20 | Sabha | Groundwater | 1853 | -7.35 | 0.02 | -44.84 | 0.11 | 13.98 | -8.09 | 0.02 | -51.17 | 0.11 | 13.52 | |
| K21 | Sabha | Groundwater | 1521 | -6.90 | 0.03 | -42.61 | 0.13 | 12.61 | -7.68 | 0.03 | -49.27 | 0.06 | 12.20 | |
| K22 | Sabha | Groundwater | 1794 | -5.91 | 0.01 | -32.95 | 0.08 | 14.33 | -7.70 | 0.02 | -47.23 | 0.10 | 14.34 | |
| K23 | Sabha | Stream | 2204 | -6.62 | 0.04 | -37.77 | 0.05 | 15.17 | -8.34 | 0.02 | -52.67 | 0.08 | 14.06 | |
| K24 | Sabha | Groundwater | 1330 | -6.08 | 0.04 | -35.28 | 0.08 | 13.40 | -7.38 | 0.02 | -45.37 | 0.11 | 13.68 | |
| K25 | Sabha | Groundwater | 1647 | -6.12 | 0.03 | -35.36 | 0.07 | 13.63 | -6.72 | 0.03 | -41.08 | 0.33 | 12.65 | |
| K26 | Sabha | Groundwater | 1330 | -7.01 | 0.02 | -41.19 | 0.04 | 14.86 | -8.00 | 0.02 | -49.24 | 0.08 | 14.80 | |
| K27 | Sabha | Groundwater | 1330 | -7.26 | 0.03 | -43.30 | 0.11 | 14.75 | -7.46 | 0.03 | -44.86 | 0.08 | 14.82 | |
| K28 | Sabha | Groundwater | 1330 | -7.12 | 0.02 | -42.96 | 0.08 | 14.01 | -8.01 | 0.03 | -49.42 | 0.08 | 14.66 | |
| K29 | Sabha | Groundwater | 1330 | -6.77 | 0.02 | -40.92 | 0.07 | 13.20 | -7.37 | 0.04 | -44.85 | 0.11 | 14.13 | |
| K30 | Sabha | Groundwater | 1330 | -7.14 | 0.03 | -42.34 | 0.07 | 14.80 | -7.48 | 0.02 | -45.32 | 0.08 | 14.55 | |
| K31 | Sabha | Groundwater | 1008 | -7.17 | 0.03 | -44.72 | 0.16 | 12.62 | -6.76 | 0.02 | -41.71 | 0.05 | 12.40 | |
| K32 | Sabha | Groundwater | 1008 | -5.71 | 0.02 | -36.63 | 0.10 | 9.06 | -5.55 | 0.02 | -35.52 | 0.06 | 8.85 | |
| K33 | Sabha | Groundwater | 1113 | -5.60 | 0.02 | -36.55 | 0.09 | 8.25 | -6.86 | 0.01 | -43.44 | 0.06 | 11.46 | |
| K34 | Barun | Snowmelt | 4071 | -6.00 | 0.02 | -29.41 | 0.09 | 18.62 | -13.94 | 0.04 | -97.25 | 0.28 | 14.30 | -100% |
| K35 | Barun | Snowmelt | 4223 | -9.82 | 0.03 | -62.41 | 0.19 | 16.13 | -12.62 | 0.05 | -86.35 | 0.15 | 14.64 | -92% |
| K36 | Barun | Stream | 5476 | -12.88 | 0.02 | -87.90 | 0.10 | 15.17 | -15.77 | 0.03 | -112.61 | 0.09 | 13.57 | -27% |
| K37 | Barun | Snowmelt | 4236 | -10.89 | 0.02 | -71.05 | 0.13 | 16.05 | -11.58 | 0.03 | -77.70 | 0.16 | 14.91 | -91% |
| K38 | Barun | Stream | 5476 | -13.41 | 0.04 | -92.46 | 0.14 | 14.80 | -15.97 | 0.03 | -114.75 | 0.09 | 13.01 | -27% |
| K39 | Barun | Snowmelt | 4367 | -8.19 | 0.02 | -48.35 | 0.15 | 17.14 | -13.20 | 0.02 | -91.59 | 0.10 | 14.05 | -95% |
| K40 | Barun | Stream | 5530 | -13.44 | 0.02 | -92.65 | 0.12 | 14.90 | -16.21 | 0.01 | -116.28 | 0.19 | 13.38 | -25% |
| K41 | Barun | Snowmelt | 4363 | -8.53 | 0.03 | -50.96 | 0.16 | 17.24 | -13.78 | 0.03 | -96.92 | 0.17 | 13.30 | -85% |
| K42 | Barun | Snowmelt | 4508 | -5.66 | 0.02 | -25.22 | 0.10 | 20.10 | -11.90 | 0.03 | -80.67 | 0.09 | 14.56 | -44% |
| K43 | Barun | Stream | 5614 | -13.73 | 0.05 | -94.63 | 0.18 | 15.18 | -16.60 | 0.03 | -119.59 | 0.24 | 13.24 | -22% |
| K46 | Barun | Stream | 5614 | -13.62 | 0.03 | -94.20 | 0.25 | 14.79 | -16.78 | 0.03 | -120.96 | 0.11 | 13.26 | -22% |
| K47 | Barun | Snowmelt | 4660 | -8.55 | 0.02 | -51.73 | 0.16 | 16.65 | -13.63 | 0.02 | -94.90 | 0.07 | 14.14 | -82% |
| K48 | Barun | Stream | 5747 | -16.42 | 0.02 | -117.68 | 0.32 | 13.66 | -17.79 | 0.03 | -129.30 | 0.13 | 13.03 | -16% |
| K49 | Barun | Glacier (debris covered) | 5248 | -10.95 | 0.03 | -72.89 | 0.17 | 14.71 | -13.94 | 0.02 | -97.69 | 0.12 | 13.84 | -49% |
| K50 | Barun | Glacier (debris covered) | 5170 | -12.26 | 0.03 | -83.37 | 0.09 | 14.70 | -14.66 | 0.04 | -103.35 | 0.07 | 13.92 | -41% |
| K51 | Barun | Glacier (pro-glacial lake) | 5728 | -17.94 | 0.03 | -130.66 | 0.16 | 12.82 | -17.70 | 0.05 | -128.83 | 0.11 | 12.81 | -17% |
| K52 | Barun | Stream | 5858 | -18.23 | 0.02 | -133.49 | 0.08 | 12.38 | -18.98 | 0.03 | -139.43 | 0.07 | 12.38 | -11% |
| K53 | Barun | Glacier (pro-glacial lake) | 5941 | -18.52 | 0.03 | -135.81 | 0.08 | 12.33 | -19.36 | 0.03 | -142.11 | 0.11 | 12.80 | -10% |
| K55 | Barun | Glacier (debris covered) | 5414 | -16.43 | 0.02 | -119.66 | 0.08 | 11.81 | -17.22 | 0.02 | -125.77 | 0.11 | 12.02 | -19% |
| K56 | Barun | Snowmelt | 4660 | -6.84 | 0.03 | -36.02 | 0.38 | 18.68 | -13.22 | 0.04 | -91.13 | 0.11 | 14.63 | -82% |
| K57 | Barun | Snowmelt | 4629 | -8.37 | 0.01 | -50.06 | 0.03 | 16.90 | -14.04 | 0.03 | -98.45 | 0.09 | 13.90 | -75% |
| K58 | Barun | Snowmelt | 4071 | -5.73 | 0.03 | -27.34 | 0.24 | 18.49 | -13.52 | 0.05 | -93.52 | 0.08 | 14.61 | -100% |
| K60 | Barun | Stream | 4757 | -9.99 | 0.03 | -64.19 | 0.11 | 15.70 | -12.04 | 0.03 | -81.55 | 0.32 | 14.76 | -47% |
| K62 | Sabha | Stream | 1536 | -5.75 | 0.02 | -32.04 | 0.16 | 13.92 | -7.48 | 0.03 | -47.28 | 0.09 | 12.59 | |
| K63 | Sabha | Groundwater | 350 | -6.12 | 0.04 | -40.63 | 0.07 | 8.30 | -6.27 | 0.03 | -41.24 | 0.05 | 8.92 | |
| K65 | Sabha | Groundwater | 350 | -5.94 | 0.04 | -38.32 | 0.09 | 9.24 | -5.96 | 0.03 | -37.69 | 0.18 | 10.02 | |

**Table 1.** δ$^{18}$O, δD, and deuterium excess values with corresponding standard deviations for each sample. Percentage change in snow is calculated based on the difference in snow extent from



Landsat 8 OLI imagery between 24 March and 11 November 2016 for the contributing drainage areas of each sample. Samples are classified based on the dominant water source type for the drainage area in the pre-monsoon season.



| Reference | Region | Season | δ18O and δD stable isotope lapse rates ($^0/_{00}$ km$^{-1}$) | Range of elevations sampled (m) | Type of surface water sampled |
|---|---|---|---|---|---|
| Mark and Mckenzie (2007) | Andes | dry season | -7 | 3600-4100 | nonglacierized springs |
| Rohrmann et al. (2014) | Andes | wet season | -0.2 to -1.7 | 340-4836 | stream |
| Varay et al. (2017) | West Himalaya | pre- and post-monsoon | -0.45 to -1.9 | 500-4200 | stream |
| Hren et al. (2009) | Himalaya | pre- and post-monsoon | -2.9 | 300-5200 | stream |
| Racoviteanu et al. (2013) | Central and East Himalaya | post-monsoon | -0.6 to -1.1 (river) -4.4 (snow) | 500-6400 5300-6400 | stream; snow |
| Wilson et al. (2016) | Central Himalaya | pre- and post-monsoon | -0.9 | 1500-4100 | stream |
| Grujic et al. (2018) | East Himalaya | pre- and post-monsoon | -2.5 | 200-3600 | stream |
| Meese et al. (2018) | East Himalaya | post-monsoon | -8.3* | 700-4900 | stream |
| van der Veen (2018) | West Himalaya | post-monsoon | -0.7 to -1.5 / -8.8* -1.6 to -3.9 / -32.7* -1.3 to -2.2 / -8.8 to -17.4* | 1000-4100 4100-5400 1000-5400 | stream |
| Voss et al. (this study) | East Himalaya | pre- and post-monsoon | -0.5 to -1.2 / -1.0 to -7.7* -2.8 to -5.8 / -23.1 to -49.2* -1.6 to -2.1 / -11.6 to -16.7* | 350-3000 4000-6000 350-6000 | stream; snow; glacier; springs |

*δD lapse rate

**Table 2.** Summary of stable isotope lapse rates and corresponding sampling approaches from other high mountain isotope studies in the Andes and Himalaya.



| Ordinary Least Squares Regression | δ18O isotope lapse rate ($^0/_{00}$ km$^{-1}$) | δD isotope lapse rate ($^0/_{00}$ km$^{-1}$) | D-excess lapse rate ($^0/_{00}$ km$^{-1}$) |
|---|---|---|---|
| Sabha Khola (<3000m) Pre-Monsoon | y = -0.5x [km] - 5.8 r$^2$=0.21; p<0.01 | y = -1.0x [km] - 39.0 r$^2$=0.01; p=0.485 | y = 3.4x [km] + 7.5 r$^2$=0.55; p<0.0001 |
| Sabha Khola (<3000m) Post-Monsoon | y= -1.2x [km] - 5.6 r$^2$=0.59; p<0.0001 | y = -7.7x [km] - 35.4 r$^2$=0.59; p<0.0001 | y = 2.2x [km] + 9.2 r$^2$=0.35; p<0.001 |
| Barun Khola (>4000m) Pre-Monsoon | y = -5.8x [km] + 17.5 r$^2$ = 0.81; p<0.0001 | y = -49.2x [km] +169.8 r$^2$=0.80; p<0.0001 | y = -2.9x [km] + 30.0 r$^2$=0.69; p<0.0001 |
| Barun Khola (>4000m) Post-Monsoon | y = -2.8x [km] - 1.0 r$^2$=0.66; p<0.0001 | y = -23.1x [km] + 9.7 r$^2$=0.66; p<0.0001 | y = -0.8x [km] + 17.9 r$^2$=0.48; p<0.001 |
| All Samples Pre-Monsoon | y = -1.6x [km] - 4.1 r$^2$=0.64; p<0.0001 | y = -11.6x [km] - 21.7 r$^2$=0.56; p<0.0001 | y = 0.9x [km] + 11.3 r$^2$=0.30; p<0.0001 |
| All Samples Post-Monsoon | y = -2.1x [km] - 4.3 r$^2$=0.95; p<0.0001 | y = -16.7x [km] - 22.4 r$^2$=0.94; p<0.0001 | y = 0.4x [km] + 11.6 r$^2$=0.20; p<0.001 |

| Weighted Least Squares Regression | δ18O isotope lapse rate ($^0/_{00}$ km$^{-1}$) | δD isotope lapse rate ($^0/_{00}$ km$^{-1}$) | D-excess lapse rate ($^0/_{00}$ km$^{-1}$) |
|---|---|---|---|
| Sabha Khola (<3000m) Pre-Monsoon | y = -0.5x [km] - 5.8 r$^2$=0.18; p=0.012 | y = -1.0x [km] - 38.9 r$^2$=0.01; p = 0.497 | y = 3.1x [km] + 8.2 r$^2$=0.59; p<0.0001 |
| Sabha Khola (<3000m) Post-Monsoon | y = -1.3x [km] - 5.6 r$^2$=0.77; p<0.0001 | y = -8.1x [km] - 35.3 r$^2$=0.67; p<0.0001 | y = 2.1x [km] +9.3 r$^2$=0.32; p<0.001 |
| Barun Khola (>4000m) Pre-Monsoon | y = -5.9x [km] +18.0 r$^2$=0.79; p<0.0001 | y = -49.8x [km] +173.3 r$^2$=0.79; p<0.0001 | y = -2.7x [km] +29.3 r$^2$=0.68; p<0.0001 |
| Barun Khola (>4000m) Post-Monsoon | y = -2.8x [km] - 0.7 r$^2$=0.64; p<0.0001 | y = -23.7x [km] +12.1 r$^2$=0.64; p<0.0001 | y = -0.9x [km] +18.2 r$^2$=0.54; p<0.0001 |
| All Samples Pre-Monsoon | y = -1.2x [km] - 4.8 r$^2$=0.69; p<0.0001 | y = -8.5x [km] - 28.0 r$^2$=0.60; p<0.0001 | y = 0.8x [km] + 11.6 r$^2$=0.24; p<0.001 |
| All Samples Post-Monsoon | y = -2.1x [km] - 4.3 r$^2$=0.95; p<0.0001 | y = -16.4x [km] - 23.0 r$^2$=0.94; p<0.0001 | y = 0.4x [km] + 12.0 r$^2$=0.14; p<0.01 |

**Table 3.** Summary of ordinary least squares and weighted regression calculations used to derive stable isotope lapse rates.