# Peer review of "Variation of deuterium excess in surface waters across a 5000-m elevation gradient in the east-central Himalaya"

_Hydrology and Earth System Sciences, 2018_

## Referee Comment (RC1) · Anonymous Referee #1 · 2 Jan 2019

The authors present the results of oxygen and hydrogen stable isotope measurements in river, ground, snow, glacier and lake waters in two river basins in the Himalaya Mountains and based on the discussion of the data, they conclude "Himalayan surface water stable isotope lapse rates strengthen in high-elevation regions". Further, several inferences are made on the importance on the contribution of different water types (snow, lake, glacier) and weather types to river discharge. The topic of water sources in Central Asia is an important one and such studies are most needed and welcomed; however the present manuscript does not fall in these categories – there are several methodological shortcomings that render the results and their interpretation, to say the least, problematic. These are detailed below. Some can be addressed, but some not

[Figure]

– the data set is all the authors had and the study must do with it. Main issues 1) Sampling – while I understand the difficulty of sampling in the mid- and high-altitude Himalayas, a well-planned field campaign could have resulted in better data. For instances, if groundwater samples were collected in SK watershed, I don't understand why river samples were not. Two maps of stable isotope distribution in groundwater and surface water would have yielded a better picture of the processes in the region. Anyway, what we have is what we have; this is merely a suggestion for future studies. 2) Amalgamation of different water types. This is my main concern – why (fig. 2) were all water types mixed in the analyses of lapse rates (at least, this is my understanding of the plots in fig. 2). Groundwater is very conservative in terms of O and H stable isotopes, while river water reflects minute changes (e.g., fig. 4 in the manuscript by Voss et al.) – as such, calculating lapse rates based on mixing all water types is wrong (were indeed glacier data included in the charts in fig. 2? The ice could be thousands of years old. . .). I suggest calculating the lapse rates separately for river and snow data. Further, I it seems that the lack of correlation in the pre-monsoon values (fig. 2) in the SK basin is an indicator of the bias towards groundwater samples, which are more conservative. In the BK basin, where mostly river samples are used, the lapse rates seem more consistent (both between them and with previously published data). 3) I am not sure I understand how the presentation of data in fig. 3 is used to understand the "elucidate the relationship between precipitation moisture source and d-excess". In the absence of moisture source determination using precipitation data (e.g., by using the HYSPLIT model) the above inference seems difficult to make. Fig. 3 seem to indicate that the stable isotope values in snowmelt change more between the pre and post monsoon period, than in glaciers (which is normal, melting of snow – X axis does not influence much the stable isotope composition of glacier-dominated rivers – as expected). 4) No information is given on discharge – this is mandatory if water resources are to be discussed. The discussion in chapter 5.2 is the most important section of the text, although it is difficult to assess its accuracy, given the mixing of data described above. Further, the conclusions seem to be a collection of generalist information, rather than the outcomes of the study. I suggest the authors 1) to separate the analysis of data based on water types, and 2) re-focus the discussion on the variability of the stable isotope values in rivers and next how are these changed by snowmelt. The discussion of seasonality seems difficult, due to the lack of data (stable isotope values in snow measured several months after snowfall (indeed, how were these samples collected – surface snow, vertical profile through the entire snowpack?) are a poor proxy for the initial value of precipitation. There are several short comments, but these could be left out now.

A word on terminology: while common in oral communication, several phrases are not accepted in written text: water isotopes do not exists, only O and H isotopes in water, "isotopically depleted snow" should be "isotopically heavy snow", "Isotopic lapse rates" do not exists and so on. Please read carefully throughout the text an correct all these errors. A good starting point for nomenclature and terminology could be Z. Sharp's "Principles…"

---

## Referee Comment (RC2) · Anonymous Referee #2 · 19 Jan 2019

Dear Editor and Authors,

please see PDF for the general and specific comments on the manuscript.

Please also note the supplement to this comment:
https://www.hydrol-earth-syst-sci-discuss.net/hess-2018-534/hess-2018-534-RC2-supplement.pdf

---

## Author Comment (AC1) · 15 Feb 2019

The comment was uploaded in the form of a supplement:
https://www.hydrol-earth-syst-sci-discuss.net/hess-2018-534/hess-2018-534-AC1-supplement.zip

---

## Author Comment (AC2) · 15 Feb 2019

**Thank you for your thoughtful comments to the manuscript. Please refer to general comments in response to Reviewer #1 for an introduction describing the broad changes made to the manuscript. Specific responses to Anonymous Review #2 are below. Figure and text updates are available in the Supplement .zip file.**

General remarks: The manuscript describes the results of a sampling campaign in the Himalayan in 2016 with the aim to determine the isotope altitude gradient for the study region. The major findings are nonlinear trends of the isotope ratio with respect to altitude, which are explained by changes in moisture sources including snowmelt (e.g. P10L41). A second emphasis is put on the deuterium excess value to delineate moisture sources.

The manuscript is adequately organized and contains no grammatical or orthography errors. Text is concise and to the point. The English is fine and does not need any revision; but note that I am not a native speaker.

However, the manuscript suffers from many technical and scientific drawbacks. For the technical details on terminology please see my comments below, but I got the impression that the field of isotope hydrology is rather new for the first author with respect to conventions and expressions. Using percentage differences in isotope values as for d in Figure 3 is useless.
**See comments below regarding the use of percentage differences in isotope values.**

From the scientific point of view, I see a major flaw in using a mixture of various water samples from snow and ice, glaciers, surface water, river water, springs and groundwater, and lakes to determine the isotope altitude gradient. Note that this gradient relates to precipitation and not to the water types sampled in this study. River water can be used as a substitute under specific circumstances but the authors fail to explain how and why their samples can be used to determine the isotope altitude effect for the region. So, how can you use a suite of various sample types to conclude on a specific parameter (altitude effect) that, in fact, relates to a complete different type of samples (rain water)?
**In response to this Reviewer's comments as well as Reviewer #1's comments, we disaggregated the data to analyze the isotope-altitude relationship for river water samples separately from samples derived predominately from snowmelt, glacial melt, and groundwater. See comments above in response to Reviewer #1.**

Overall, the study also lacks clear scientific objectives in the introduction. Some ideas are mentioned but it remains unclear (also in the conclusions) if the results could be used to e.g. better manage high mountain water resources on a local (see e.g. Königer et al. 2017, doi:10.1002/hyp.11291) or regional scale.

The authors need to clearly explain what were the aims of the study (Introduction), how does the sampling strategy was designed to fulfill these scientific goals (Methods) and how does the data (Results) helped to solve these scientific question (Discussion and Conclusions; the latter with an outlook to future work and recommendations).
**Please see our overview response (point #3) as well as our reply to Review #1's fourth comment describing how the manuscript text was edited to distill the narrative in our introduction, methods, results, and discussion to highlight the application of oxygen and**

**hydrogen isotopes (calculated as lapse rates and as a tracer utilizing deuterium excess) to investigate the influence of snow and glacial melt waters on river water composition.**

Unfortunately, I have to reject the study for publication in HESS at current stage but encourage the authors to re-submit a strongly revised version of this interesting data set.

Specific comments:
P2L14: The definition of deuterium excess is a bit awkward. I notice what you mean, but in fact deuterium excess itself is not a "deviation from the GMWL" but its definition, of course, used the dual-isotope equation of the GMWL. By definition, $d$ is calculated with the slope of 8 from the GMWL and can be visualized as the intercept of a line with slope 8 which crosses the given isotope pair in a dual isotope plot of $\delta^{18}O$ vs $\delta^{2}H$. Thus, the GMWL connects points of $d = 10$ ‰. Either rephrase here or simply delete "which….(GMWL)," to avoid confusion. Also note that sea surface temperature (SST) also modifies the deuterium excess. Please this information to the sentence.
**Thank you for this recommendation, the phrasing "as a deviation from the GMWL" has been removed to improve clarity. We included text and a citation to highlight that sea surface temperature also modifies deuterium excess (P2L13-19).**

P1L39: The term stable isotope "lapse rates" is unfamiliar to me. In hydrology, to my knowledge, this is typically referred to as '(isotope) altitude effect' or 'altitudinal (isotope) gradient' (cf. Mook 2000) Mook, W.G. (ed.) (2000): Environmental Isotopes in the Hydrological Cycle - Principles and Applications . - International Hydrological Programme (IHP-V), Technical Documents in Hydrology, 39 , IAEA/UNESCO.
**The term "lapse rate" is common to describe the relationship between oxygen and hydrogen isotopes as they relate to increasing altitude. Poage and Chamberlain (2001)'s review article utilizes the term throughout the paper and highlights numerous other studies that use the same terminology.**

P2L16: In addition, the information that $d$ relates to the moisture source (and more specifically to the rH and T at this point of origin during evaporation) is missing from the text. This makes this sentence hard to understand for the readers as you here directly move on to the identification of sources of precipitation by deuterium excess values. Please provide here a few more details.
**We added additional context to transition from our broad introduction of deuterium excess to how it can be used specifically in the Himalayan context (P2L17-22).**

P2L38: "Arun River catchment" (catchment/watershed/basin missing from text)
**Updated to include "catchment" (P2L41).**

P3L1: change to '8480 m in altitude' to make clear that you are talking about height (and not distances) here.
**Changed to "8480 m in altitude" (P2L45).**

P4L10-25 and Figure 1: It is unclear to me how groundwater (Figure 1) relates to the type of samples described in the text. I assume that the term "springs and surfacewater tributaries […] (N = 50)" relates to these points? Please bring text and figure labels/caption in line so that the

reader can follow what samples are taken at which location.  Also note that lakes might undergo strong evaporation processes, also in high and cold environments and that these samples might be not indicative of the original precipitation.  Overall, samples from river water are rather problematic with respect to the altitude effect as river water is affected by the 'catchment effect' (see Dutton et al. 2005). High altitude springs, gaining groundwater conditions along the river course and surface runoff change the isotope values with respect to the on-site precipitation. This can be of minor influence, but for rivers in high altitude regions this should definitely be considered and discussed. Dutton, A., Wilkinson, B.H., Welker, J.M., Bowen, G.J. and Lohmann, K.C. (2005): Spatial distribution and seasonal variation in 18O/16O of modern precipitation and river water across the conterminous USA. - Hydrological Processes , 19 , 4121-4146.

**We added a specific description of the groundwater samples (P5L29-32).  Additionally, we provided a new text regarding the "catchment effect" as it relates to our river $\delta^{18}$O and $\delta$D lapse rates (P3L11-15; P15L39-41). Our updated results, with the partitioning of $\delta^{18}$O and $\delta$D lapse rates by water source/type, further elucidate the discussion offered in Dutton et al. (2005).  In the article, Dutton et al. allude to the potential non-linearity of $\delta^{18}$O and $\delta$D lapse rates in river water draining from high elevation catchments, but they did not have the data to test this theory.  Our results validate their hypothesis and offer additional insights regarding the "catchment effect" as river water mixes various water sources from upstream portions of the catchment.**

P4L35:  Either 'L-2140i cavity ringdown' or 'laser spectrometer'
**This has been changed to Picarro L-2140i Laser Spectrometer (P5L46).**

P4L38:  round hydrogen isotope precision to 0.3‰
**This is changed to 0.3 ‰.**

P4L38:  This sentence makes no sense. Stable isotope analytical uncertainty can never be reported in percent, as it is not a concentration but a relative deviation. Therefore, standard deviations must always be given in the same unit as the value (in this case in permil that serves as a unit).  Change to 'Analytical uncertainties for the stable isotope are reported as standard deviation in the data repository.'  Note also that the term „water stable isotope" does not exist. Water has no stable isotopes but oxygen and hydrogen of the water molecule does.
**The text has been changed accordingly with additional details to the standard deviation calculated from triplicate measurements (P6L1-3).  Terminology has been updated throughout the paper to remove the use of "water stable isotope" and substitute with oxygen and hydrogen isotopes or similar.**

P4L40/41: See my comment above about deuterium excess. Simply indicate that you have calculated the *d*-value after that equation. I also suggest to give the equation clearly in a single line with number and change the asterisk to 'Å~'.
**Changes have been made to not refer to d-excess as a deviation from the GMWL, though we kept the equation in the text rather than separating it into its own line.**

P5L2:  see my comment above. Change to "4.1 Stable isotope values…"; next line: "oxygen and hydrogen isotope values…"

**This heading now reads: "4.1 Oxygen and hydrogen isotope values and lapse rates" (P6L24).**

P5L29: "r2 of 0.01" makes no sense to me. Looking on Figure 2b I cannot see how the r2 value of the linear regression of the Sabha Khola Pre-Monsoon can be 0.01? By simply looking on the data this must be higher (or better) from my impression. Or not all data points are shown. Please check and correct.

**We completed new regression analysis of the oxygen and hydrogen isotope – altitude relationship for the various river, snowmelt, glacial melt, and groundwater sources. All regression results are updated accordingly.**

P6, Figure 2: This Figure shows the results of the study. However, it is unclear to me what is exactly shown. First, have all water types been mixed (river, snow, groundwater, spring, lakes)? As each of these hydrological compartments have their own behavior with respect to isotopic fractionation, it makes not really sense to mix them up (or at least not to use different symbols to separate them). Second, what is shown on the x-axis? Is this the sampling site altitude or (as stated in the caption) the "mean catchment elevation of the drainage areas of the samples". If the latter – how was that elevations derived? From a GIS model of the sub-catchments? Please clarify here, not only in the Figure caption bus also in the text. Third, the rather stable isotope values in the lower regions of the watershed (<3000masl) could be the result of a sampling artifact as most samples of the lower region seems to be influences by groundwater (Figure 1b, Sabha Khola), which tend to have rather stable isotope values with low seasonality while the steeper gradients relies on samples that were sampled close or along the river course (Figure 1a).

**Our analysis of the $\delta^{18}$O and $\delta$D lapse rates is now updated (see response to Reviewer #1) and partitions $\delta^{18}$O and $\delta$D lapse rates by the dominant water sample type (river, snowmelt, glacial melt, groundwater springs/wells, groundwater-sourced surface water tributaries). The x-axis is the second interpretation: the mean catchment elevation of the drainage area of the samples. We edited the figure caption and provide a description of how the mean catchment elevations of the drainage area of the samples were calculated in the methods section (P6L5-9). New discussion has been added to the manuscript regarding the mixing of water sources (snowmelt, glacial melt, groundwater) in the river samples and the mixing/catchment effect on $\delta^{18}$O and $\delta$D lapse rates (see section 5.1).**

P7L5: See my comment on precision of stable isotope values. Simply state that error is smaller than symbol size and skip percent values here.

**Figure captions and text are revised accordingly.**

P7L22: What is the 'anomaly'? Do you refer the sample in Figure 3 that plots closer to the glacier-melt dominated regions (x-axis)? Please clarify.

**We removed this text as the discussion does not add to or detract from the figure interpretation.**

Figure 3: As already mentioned above, I don't think that percentage can be used here to express differences on *d*-values on the y-axis. Use Δ*d* values here (the difference in deuterium excess between pre and post monsoon) and not percent values, which makes no sense here.

**Thank you for your concern regarding the use of percentages to compare isotope values**

across seasons; however, we believe that the use of a percent is a useful metric to look at the proportional change in the d-excess and snow extent data across seasons. For example, an absolute change in d-excess of $2^0/_{00}$ is a much more significant change if the sample's value changed from $1^0/_{00}$ in the pre-monsoon to $3^0/_{00}$ in the post monsoon instead of $20^0/_{00}$ to $18^0/_{00}$. We are interested in the magnitude of change across seasons. That said, the absolute difference in d-excess produces the same overall patterns with greater loss of snow correlating to greater differences (decreases) in d-excess values across seasons (see figure here).

[Figure]

P10L19/20: You have to consider that the sampling of river water always gives you an integrated signal from the regions above your sampling point as tributaries are continuously entering your stream (in contrast to precipitation samples). As a consequence you will never measure the real (or true) "precipitation isotope signal" in a river, especially not in high relief terrains as in your study. Thus, you cannot use a river water sample to calculate the (precipitation) elevation isotope gradient. Your river water signal will be always more negative than any on-site precipitation as the river water represents a mixture of high altitude water and water derived from sources near your the sampling site. This may led to a significant underestimation of the real isotope altitudinal gradient with increasing uncertainty further downstream. Please include a brief discussion about the limitations of river water samples with respect to rainfall-derived lapse rates. I agree that river water data can be used to estimate on altitudinal isotope gradients but be careful to clearly state that your values relate to river water (and not to rainfall).

We agree with Reviewer #2 that the mixing of different sources into river water samples is a critical control on the $\delta^{18}O$ and $\delta D$ lapse rates as the river water samples represent an

**integrated value of source water inputs upstream of the collection point. As previously mentioned, we include new discussion regarding this "catchment effect" on our results and have modified the text to clearly state that we are interested in calculating oxygen and hydrogen stable isotope lapse rates for surface waters rather than a pure precipitation-derived lapse rates.**

P16L16: I doubt that you can cite a manuscript "under review" (van der Veen). Check with the Editor but either it is published at time of final acceptance of this manuscript or delete.
**Based on the HESS references guidelines it appears that "in review" papers can be cited, though this citation will be updated as recommended by the editor or if the paper is accepted by the time of publication (https://www.hydrology-and-earth-system-sciences.net/for_authors/manuscript_preparation.html)**

Technical comments
P1L16 and throughout the manuscript: please change $\delta$D to $\delta^2$H (note: delta-symbol italic) to follow the actual conventions for the notation of stable isotopes (Coplen 2011; Brand et. al. 2014) Brand, W., A., Coplen, T., B., Vogl, J., Rosner, M. and Prohaska, T. (2014): Assessment of international reference materials for isotope-ratio analysis (IUPAC Technical Report). - Pure and Applied Chemistry , 86 , 263-467, [doi:10.1515/pac-2013-1023]. Coplen, T.B. (2011): Guidelines and recommended terms for expression of stable-isotope-ratio and gas-ratio measurement results. - Rapid Communications in Mass Spectrometry, 25, 2538-2560, [doi:10.1002/rcm.5129].
**Delta-symbol is now in italics throughout the manuscript.**

P2L14 and throughout the manuscript: Change the '*d*' to italic: "Deuterium excess (dexcess)"; or to '*d*' only (not '-excess' after *d*) but I notice that both abbreviations are used in the literature.
**It appears that d-excess is a commonly used and accepted term to describe deuterium excess as is "*d*" or simply "d" (see Gat (2010)). We kept our terminology as d-excess or deuterium excess in the text.**

P2L44: introduce abbreviation 'masl'
**Meters above sea level (masl) is now introduced (P3L2).**

P4L19: space characters missing (N=6 -> N = 6); check also line 24.
**Spaces were added accordingly.**

Table 1: In headline change to permil-symbol (‰); not 0/00. To what does the "std. dev. for each sample" refer to? To the number of injections in the laser instrument? Then specify the number of injections in Table caption and methods. Also check superscript of 18O (also in in Tables 2 and 3). Further: The samples have no date/time? Please add this information as it essential to the reader; in particular for the river water samples taken from the streams. Please also provide a table with the exact GPS locations for each sample in the supplementary material (or include here). The sampling locations cannot be identified from Figure 1 and need further details provided either in the tables or in the supplementary material.
**Table 1 is now updated with the permil-symbol and $^{18}$O correctly displayed and includes all samples' latitude, longitude, date, and time details related to the sample collection. The**

**standard deviation calculation is now described in the methods section (P6L1-3) as well as in the caption of the table.**

Table 3: What was the weighing parameter for the regression equation? And why was it not used or discussed in the text? Please specify.
**The regression results reported throughout the text use ordinary least squares and the weighted regression results that are included supplementary in Table 3 use a bisquare weight parameter.**

Figure 2: Lines in figure appear grey in my copy. I suggest using clear black and white colors for axis, lines and also symbols (no grey filling, make them black and put the white symbols on top layer) In (c) change y-axis label to 'd-excess' (not D-Excess). Also for Figure 5.
**Figures are now adjusted to be solely black and white.  Figure axis labels for previous Figure2(c) (now Figure 4) and Figure 5 (now Figure 6) are updated to read "Deuterium excess" rather than "D-excess".**

Figure 3: y-axis makes no sense. See my comments above.
**See reply above.**

Figure 4: Change rainfall data to bar graphs with suitable resolution. The 'spikes' does not give a correct impression about the rainfall data collection interval (daily?). Same for the weekly T data.
**Rainfall data are now displayed as bar graphs.**

**References cited in the response:**
**Dutton, Andrea, Bruce H. Wilkinson, Jeffrey M. Welker, Gabriel J. Bowen, and Kyger C. Lohmann. 2005. "Spatial Distribution and Seasonal Variation in 18O/16O of Modern Precipitation and River Water across the Conterminous USA." *Hydrological Processes* 19 (20): 4121–46. https://doi.org/10.1002/hyp.5876.**
**Gat, Joel R. 2010. *Isotope Hydrology: A Study of the Water Cycle*. Vol. 6. Series on Environmental Science and Management. PUBLISHED BY IMPERIAL COLLEGE PRESS AND DISTRIBUTED BY WORLD SCIENTIFIC PUBLISHING CO. http://www.worldscientific.com/worldscibooks/10.1142/p027.**
**Poage, Michael A., and C. Page Chamberlain. 2001. "Empirical Relationships Between Elevation and the Stable Isotope Composition of Precipitation and Surface Waters: Considerations for Studies of Paleoelevation Change." *American Journal of Science* 301 (1): 1–15. https://doi.org/10.2475/ajs.301.1.1.**